# Intravenous immunoglobulin treatment in childhood encephalitis (IgNiTE): a randomised controlled trial

Matilda Hill ![ORCID] ,[1] Mildred Iro,[1] Manish Sadarangani,[1,2,3,4] Michael Absoud,[5,6] Liberty Cantrell,[1] Kling Chong,[7] Christopher Clark,[7] Ava Easton,[8,9] Victoria Gray,[10] Rachel Kneen,[11,12] Ming Lim,[5,6] Xinxue Liu ![ORCID] ,[1] Michael Pike,[13] Tom Solomon,[12,14,15,16] Angela Vincent,[17,18] Louise Willis,[1] Ly-Mee Yu,[19] Andrew J Pollard ![ORCID] ,[1,2] The IgNiTE study team

MH and MI contributed equally.

For numbered affiliations see end of article.

**Correspondence to**
Dr Matilda Hill;
matilda.hill@paediatrics.ox.ac.uk

## ABSTRACT

**Objective** To investigate whether intravenous immunoglobulin (IVIG) improves neurological outcomes in children with encephalitis when administered early in the illness.

**Design** Phase 3b multicentre, double-blind, randomised placebo-controlled trial.

**Setting** Twenty-one hospitals in the UK.

**Participants** Children aged 6 months to 16 years with a diagnosis of acute or subacute encephalitis, with a planned sample size of 308.

**Intervention** Two doses (1 g/kg/dose) of either IVIG or matching placebo given 24–36 hours apart, in addition to standard treatment.

**Main outcome measure** The primary outcome was a 'good recovery' at 12 months after randomisation, defined as a score of ≤2 on the Paediatric Glasgow Outcome Score Extended.

**Secondary outcome measures** The secondary outcomes were clinical, neurological, neuroimaging and neuropsychological results, identification of the proportion of children with immune-mediated encephalitis, and IVIG safety data.

**Results** 18 participants were recruited from 12 hospitals and randomised to receive either IVIG (n=10) or placebo (n=8) between 23 December 2015 and 26 September 2017. The study was terminated early following withdrawal of funding due to slower than anticipated recruitment, and therefore did not reach the predetermined sample size required to achieve the primary study objective; thus, the results are descriptive. At 12 months after randomisation, 9 of the 18 participants (IVIG n=5/10 (50%), placebo n=4/8 (50%)) made a good recovery and 5 participants (IVIG n=3/10 (30%), placebo n=2/8 (25%)) made a poor recovery. Three participants (IVIG n=1/10 (10%), placebo n=2/8 (25%)) had a new diagnosis of epilepsy during the study period. Two participants were found to have specific autoantibodies associated with autoimmune encephalitis. No serious adverse events were reported in participants receiving IVIG.

**Conclusions** The IgNiTE (ImmunoglobuliN in the Treatment of Encephalitis) study findings support existing evidence of poor neurological outcomes in children with encephalitis. However, the study was halted prematurely and was therefore underpowered to evaluate the effect of

early IVIG treatment compared with placebo in childhood encephalitis.

**Trial registration number** Clinical Trials.gov NCT02308982; ICRCTN registry ISRCTN15791925.

### STRENGTHS AND LIMITATIONS OF THIS STUDY

⇒ This was the first ever multicentre, randomised controlled trial evaluating intravenous immunoglobulin treatment for all-cause encephalitis in children.

⇒ The study had clinically meaningful endpoints and was run to a very high standard, with rigorous blinding procedures throughout.

⇒ Recruitment to the study was limited by the strict inclusion and exclusion criteria, the limited time window for enrolment and lack of equipoise among clinicians.

## INTRODUCTION

Encephalitis is a major cause of illness and death globally.[1–3] It is characterised by inflammation of the brain parenchyma causing neurological dysfunction which manifests acutely as altered mental state and can have long-term sequalae including neurological disability and seizures. In children, the most common causes of encephalitis are autoimmune disorders and infections, with viral encephalitis being the most frequently identified aetiology.[4–10] It often takes time to reach a definitive diagnosis, and a cause may not be found despite extensive investigation in at least one-fifth of children.[4 6 7 10]

Encephalitis is more prevalent among children than adults, with an estimated incidence of 4.0–12.6 per 100 000 person years for children in high-income countries.[7 10–14] There is a substantially higher burden of childhood encephalitis in regions such as southeast Asia where the Japanese encephalitis virus is endemic.[2 8] Childhood encephalitis carries

a significant mortality rate; this ranges from 5% to 13%, dependent on setting and aetiology.[4 8 10 15 16] Approximately half of children who survive an episode of encephalitis will have long-term sequalae which may include neurological deficits, physical disability, cognitive impairment, neuropsychiatric disorders and epilepsy.[4 8 10 15 17–19] Childhood encephalitis is therefore associated with a high global economic, healthcare and social burden.[1 3 8 15 20]

While there is good evidence for the efficacy of aciclovir in the management of encephalitis caused by herpes simplex virus and varicella zoster virus,[21 22] there are limited therapeutic options for other types of childhood encephalitis and the mainstay of treatment is supportive care. Treatment strategies for autoimmune encephalitis include methylprednisolone, plasma exchange and intravenous immunoglobulin (IVIG), but the recommendations for their use are based largely on studies in individuals with specific types of autoimmune encephalitis, retrospective cohort studies and expert opinion.[23–26] Furthermore, these therapies are often only implemented after a definitive autoimmune cause for encephalitis has been identified or all alternative diagnoses, including infectious, have been ruled out.

IVIG is used successfully in other inflammatory and neurological conditions in children[27 28]; however, there have been no high-quality studies to support or refute its use in children with all types of encephalitis.[29 30] Inflammation of the brain parenchyma is the common cause of altered neurological function in encephalitis, regardless of the aetiology, and it may therefore be postulated that interventions which attenuate the inflammation early in the illness are likely to have the greatest efficacy in reducing the severity of the acute illness, mortality and neurological sequalae of childhood encephalitis.

In this study, we set out to establish if early IVIG treatment, in addition to standard care, improves outcomes for children with encephalitis of all aetiologies.

## METHODS

### Study design

IgNiTE was a randomised, double blinded, parallel arm, placebo-controlled study to compare early IVIG treatment with placebo in the treatment of childhood encephalitis in individuals aged 6 months to 16 years. It was conducted across 21 National Health Service (NHS) hospitals in the UK. Participants were followed up for 12 months after randomisation, with outcomes assessed during the acute admission, at 4–8 weeks after discharge from acute care, at 6 months after randomisation, and 12 months after randomisation.

The trial was prospectively registered with Clinical-Trials.gov (identifier NCT 02308982) on 5 December 2014. The trial was assigned an International Standard Randomised Controlled Trial Number on 24 June 2015 (ISRCTN 15791925), and a European Clinical Trials Database number (2014-002997-35). A Trial Steering Committee (TSC) was established to oversee the trial, and

an independent Data Monitoring and Ethics Committee was set up to monitor the safety, efficacy and overall conduct of the study.

The original trial protocol was published on 3 November 2016.[31] The protocol was amended after the early termination of the trial to remove endpoints which could not be derived from the data collected and to update the statistical analysis section; the amended protocol is available in the online supplemental material.

### Participants

Eligible participants were hospitalised children aged between 6 weeks and 16 years who met the case definition for encephalitis based on the consensus definition by the International Encephalitis Consortium,[32] where written informed consent was obtained from parents or guardians, and assent was given if appropriate.

Exclusion criteria were a high clinical suspicion of bacterial meningitis; prior receipt of IVIG during the admission or known contraindication to IVIG; traumatic brain injury; history of metabolic encephalopathy; stroke, toxic or hypertensive encephalopathy; pre-existing demyelinating disorder; significant renal impairment; hypercoagulable state; hyperprolinaemia; participation in another research trial involving an immunomodulatory treatment; pregnancy; any significant disease or disorder which may put the participants at risk because of participation in the trial, influence the result of the trial, or the participant's ability to participate in the trial; involvement in another research trial involving an investigational medicinal product (IMP) which has potential immunomodulatory or neuroprotective effects.

### Intervention

Two doses of 1 g/kg/dose of either IVIG or a matching volume of placebo were given 24–36 hours apart, with the first dose administered as soon as possible after enrolment and within five working days from the suspicion of an encephalitis diagnosis.

The active treatment (IVIG) used in the study was privigen (100 mg/mL solution), manufactured and provided by CSL Behring. The placebo was a mixture of 0.9% saline and 0.1% human albumin solution, manufactured at the Royal Broadgreen and Liverpool Aseptic Production Unit, Liverpool, UK under cGMP conditions and its Manufacturer's Importer's Authorisation (IMP) licence.

### Randomisation and blinding

Participants were randomised 1:1 to IVIG or placebo treatment after consent was obtained. Randomisation was stratified by age group (< 1 year, 1–4 years, 5–9 years, 10–14 years and ≥15 years) and steroid treatment at the time of randomisation, using stratified block randomisation with randomly varying block sizes. Randomisation was performed using a secure web-based randomisation system (Sortition) which was developed by the Clinical Trials Unit in the Nuffield Department of Primary Care Health Sciences, University of Oxford.

Participants, their parents or guardians, clinical staff and all study staff (including staff involved in recruitment, administration of study treatment, data collection and entry, and laboratory analyses) were blind to the treatment allocation through the entire study period. Study monitors who were independent of the study and all site pharmacists were unblinded to ensure dispensing of the correct allocation and robust IMP management at each study site. The placebo and IVIG were visually identical, due to the additional of 0.1% human albumin solution to 0.9% in the placebo.

## Primary outcome

The primary outcome was good recovery, which was defined as a score of 2 or less on the paediatric version of the Glasgow Outcome Score Extended (GOS-E Peds) at 12 months after randomisation.

The GOS-E Peds is based on the GOS-E, a gold standard for measuring outcomes in adults with traumatic brain injury. It is a validated tool for use in children, and provides a developmentally appropriate structured interview necessary to evaluate children across different age groups.[33] Participants were assigned a GOS-E Peds score: 1—Upper Good Recovery, 2—Lower Good Recovery, 3—Upper Moderate Disability, 4—Lower Moderate Disability, 5—Upper Severe Disability, 6—Lower Severe Disability, 7—Vegetative State, and 8—Death. 'Good recovery' was defined as a GOS-E Peds score of ≤2, and a score of >2 indicated 'poor recovery'.

## Secondary outcomes

Secondary clinical outcomes included admission to intensive care unit, requirement for invasive ventilation, length of acute hospital stay, new diagnoses of epilepsy and need for antiepileptic treatment in the 12 months after randomisation.

Secondary neurological and functional outcomes comprised GOS-E Peds assessment at 6 months after randomisation, and Liverpool Outcome Score (LOS) assessment, Pediatric Quality of Life Score (PedsQL) assessment, Gross Motor Function and Classification System (GMFCS) assessment, Strengths and Difficulty Questionnaire (SDQ) assessment and Adaptive Behavior Assessment System—second edition (ABAS-II) assessment at 4–8 weeks after discharge from acute care and at 12 months after randomisation.

Secondary neuropsychological outcomes were cognitive assessment at 12 months after randomisation using the age-appropriate scales: (1) Bayley Scales of Infant and Toddler Development, third edition (1 to 2 years 5 months); (2) Wechsler Preschool Primary Scale of Intelligence IV (2 years 6 months to 5 years 11 months), and (3) Wechsler Intelligence Scale for Children V (6 years to 16 years 11 months).

The secondary neuroimmunology outcome was identification of autoantibodies. The antibodies tested for were for antibodies against live neurons, aquaporin 4, N-methyl-D-aspartate receptor, myelin oligodendrocyte glycoprotein (MOG), leucine-rich, glioma inactivated 1 (LGI1), and contactin-associated protein-like 2.

Secondary neuroimaging outcomes comprised assessment of CT or MRI brain scans performed as part of routine care during the acute illness, and follow-up scans performed at 6 months after randomisation in a subset of participants, where consent was provided.

Secondary safety outcomes included safety data obtained throughout the study, and a full blood count performed for all participants 24–48 hours following the second dose of the study treatment to monitor for haemolysis which has previously been described with high concentrations of IVIG treatment.[34] Safety data comprised adverse events (AEs) and adverse events of special interest occurring in the first 5 days following receipt of each dose of the study drug, serious adverse events (SAEs) occurring up until 6 months after randomisation and serious adverse reactions occurring throughout the study period. Information on any deaths occurring up to 12 months after randomisation was also collected.

Further information regarding to the secondary outcomes is provided in the online supplemental material.

## Protocol amendments

The IgNiTE study was halted in October 2017 after the withdrawal of funding due to slower than anticipated recruitment. This was despite the proposal of alternative strategies to deliver on the study objectives, including revision of the recruitment timeline to ensure that the objectives of this important clinical study could be met. Where possible, follow-up activities were completed for all participants who were already enrolled into the trial, as per the protocol. The protocol was amended to remove endpoints which could not be derived from the data collected and to update the statistical analysis section.

## Statistical analysis

A sample size of 308 participants recruited over a 24-month period (154 per group, with approximate 10% attrition rate) was planned to achieve 90% power (at 5% level of significance) to detect at least a 20% clinically significant treatment difference from 43% in the 'good recovery' rate (defined as a GOS-E Peds score of ≤2) by 12 months after randomisation. This was based on the results of a large observational study on autoimmune encephalitis.[26]

At the time the trial was halted, only 18 participants had been recruited. The trial was therefore underpowered to perform hypothesis testing of outcomes, subgroup comparisons or sensitivity analyses. Therefore, all analyses performed were descriptive. The analyses were performed on the intention-to-treat population; this included all 18 participants who were randomised. In the analysis of the AEs, the population analysed were the 16 participants who received study treatment.

## Patient and public involvement (PPI)

The Encephalitis Society was involved in the planning of this study, and the training of research nurses and study

recruiters. A representative of The Encephalitis Society was on the Trial Management Group and provided a patient-centred research perspective to the study design and conduct. PPI groups were consulted in the development of the essential documents for the study including the participant information sheet and consent forms. Three PPI representatives with previous personal experiences of encephalitis sat on the TSC and contributed to providing overall oversight of the study. Study update meetings were held to which patients previously affected by encephalitis were invited to share their experiences with the study team.

## RESULTS
### Participants
A total of 884 patients were screened for eligibility between 23 December 2015 and 26 September 2017 across 21 NHS hospitals, of whom 18 participants were enrolled and randomised across 12 hospital. 10 participants were

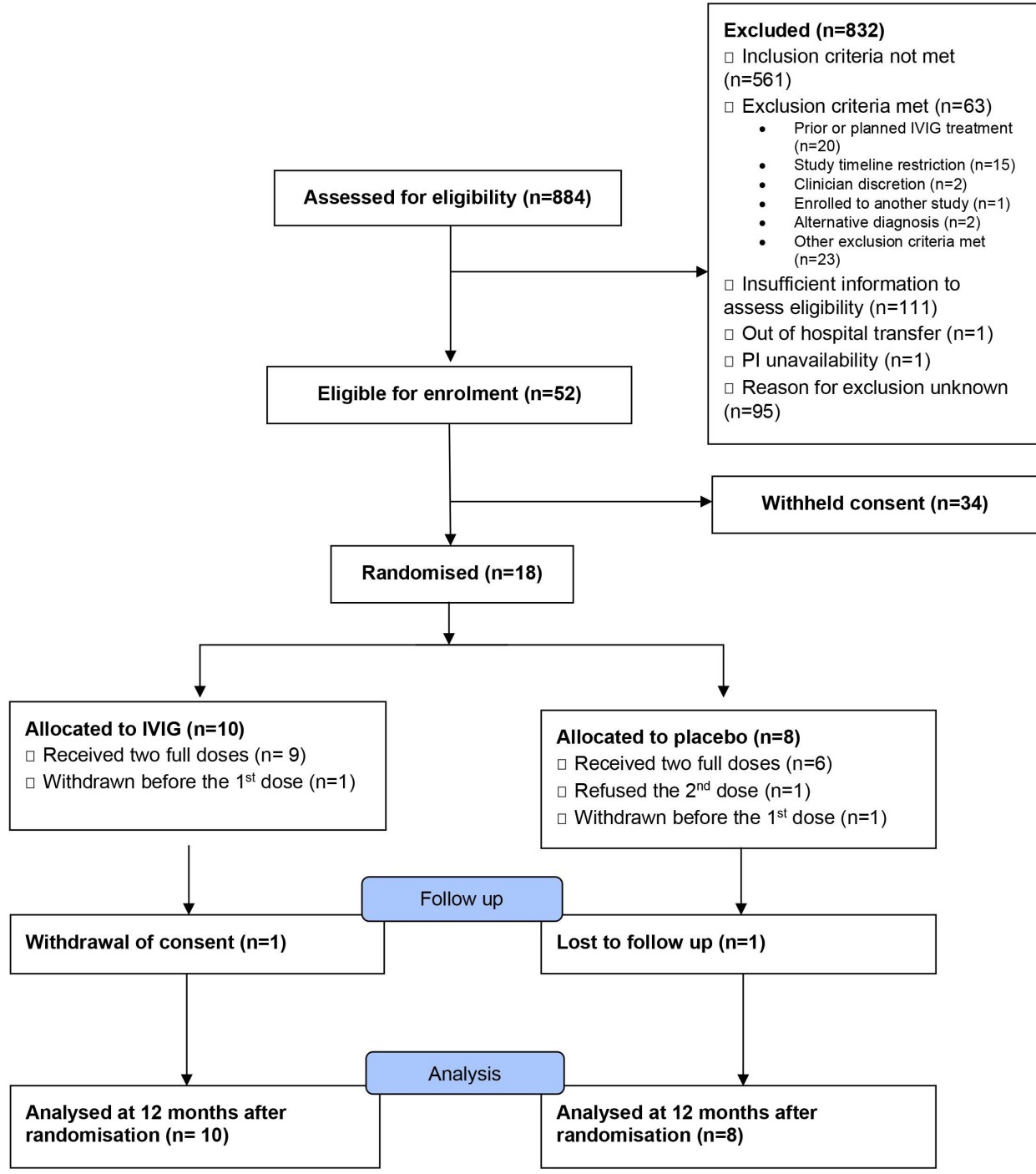

**Figure 1** Study flow diagram. IVIG, intravenous immunoglobulin.

**Table 1** Baseline characteristics of enrolled participants

| Baseline characteristic | | IVIG (n=10) | Placebo (n=8) | All (n=18) |
|---|---|---|---|---|
| Age at randomisation (years) | Median (IQR) | 5.55 (1.52–11.8) | 4.09 (2.71–9.64) | 4.09 (2.0–11.8) |
| Sex | Male | 4 (40%) | 4 (50%) | 8 (44.4%) |
| | Female | 6 (60%) | 4 (50%) | 10 (55.6%) |
| Ethnicity | White | 8 (80%) | 8 (100%) | 16 (88.9%) |
| | Asian | 1 (10%) | 0 (0%) | 1 (5.6%) |
| | Missing | 1 (10%) | 0 (0%) | 1 (5.6%) |
| History of immunocompromise | No | 9 (90%) | 7 (87.5%) | 16 (88.9%) |
| | Missing | 1 (10%) | 1 (12.5%) | 2 (11.1%) |
| Previous diagnosis of encephalitis | No | 9 (90%) | 7 (87.5%) | 16 (88.9%) |
| | Missing | 1 (10%) | 1 (12.5%) | 2 (11.1%) |
| History of encephalopathic illness | No | 9 (90%) | 7 (87.5%) | 16 (88.9%) |
| | Missing | 1 (10%) | 1 (12.5%) | 2 (11.1%) |
| Pre-existing diagnosis of epilepsy | No | 9 (90%) | 7 (87.5%) | 16 (88.9%) |
| | Missing | 1 (10%) | 1 (12.5%) | 2 (11.1%) |

assigned to IVIG treatment, and 8 patients were assigned to placebo. The study flow diagram is shown in figure 1.

Table 1 summarises the baseline characteristics of participants by treatment arm. The mean age of the participants was 4.09 years (IQR 2.0–11.8), 44% were male, and 89% were of white ethnicity.

### Primary outcome

At 12 months after randomisation, nine participants (50%; IVIG n=5/10 (50%); placebo n=4/8 (50%)) made a good recovery, defined as a GOS-E Peds score of ≤2. Five participants (28%; IVIG n=3/10 (30%), placebo n=2/8 (25%)) made a poor recovery, defined as a GOS-E Peds score of >2. Four participants (22%; IVIG n=2/10 (20%), placebo n=2/8 (25%)) did not undergo a GOS-E Peds assessment at this timepoint. Table 2 displays these results.

### Secondary outcomes

### Clinical outcomes

Ten participants (56%; IVIG n=5/10 (50%), placebo=5/8 (63%)) were admitted to intensive care during their acute admission with encephalitis, as shown in table 3. Nine of these participants (90%; IVIG n=4/5 (80%),

placebo n=5/5 (100%)) required invasive ventilation, for a median duration of 2 days (IQR 2.0–3.0). The median length of stay on intensive care was 4.5 days (IQR 3.0–6.8) and the overall median length of hospitalisation for acute care was 11 days (IQR 7.8–19.5).

Three participants (17%; IVIG n=1/10 (10%), placebo n=2/8 (25%)) had a new diagnosis of epilepsy during the study period. Five participants (28%; IVIG n=2/10 (20%), placebo n=3/8 (38%)) had incomplete data for this outcome.

### GOS-E Peds assessment at 6 months

Fifteen participants underwent GOS-E Peds assessment at 6 months after randomisation. Eight participants (44%; IVIG n=4/10 (40%), placebo n=4/8 (50%)) made a good recovery and seven participants (39%; IVIG n=4/10 (50%), placebo n=3/8 (38%)) made a poor recovery at this timepoint, as shown in table 4.

### Liverpool Outcome Score

Fifteen participants had an LOS assessment at 4–8 weeks after discharge from acute care. Five participants (28%; IVIG n=3/10 (30%), placebo n=2/8 (25%)) made a full

**Table 2** GOS-E Peds scores at 12 months after randomisation

| | IVIG (n=10) | Placebo (n=8) | Overall (n=18) |
|---|---|---|---|
| GOSE-Peds Score | | | |
| Upper good recovery | 4 (40%) | 4 (50%) | 8 (44%) |
| Lower good recovery | 1 (10%) | 0 (0%) | 1 (6%) |
| Upper severe disability | 1 (10%) | 1 (13%) | 2 (11) |
| Lower severe disability | 2 (20%) | 1 (13%) | 3 (17%) |
| Participants with missing data due to being withdrawn or lost to follow-up | 2 (20%) | 2 (25%) | 4 (22%) |

GOSE-Peds, Paediatric Glasgow Outcome Score Extended; IVIG, intravenous immunoglobulin.

**Table 3**  Secondary clinical outcomes

| Outcome | | IVIG (n=10) | Placebo (n=8) | Overall (n=18) |
|---|---|---|---|---|
| **During hospital stay** | | | | |
| Duration of ventilation | Median (IQR) | 2.5 (2.0–3.5) (n=4) | 2.0 (2.0–3.0) (n=5) | 2.0 (2.0–3.0) (n=9) |
| Length of ICU stay | Median (IQR) | 4.0 (3.0–6.0) (n=5) | 5.0 (2.0–10.0) (n=5) | 4.5 (3.0–6.8) (n=10) |
| Length of hospitalisation for acute care | Median (IQR) | 12.0 (8.0–27.0) (n=9) | 8.0 (6.5–14.0) (n=7) | 11.0 (7.8–19.5) (n=16) |
| **6 months post randomisation** | | | | |
| New diagnosis of epilepsy since discharge | n (%) | 1 (10%) | 1 (13%) | 2 (11%) |
| Anti-epileptic treatment since discharge | n (%) | 1 (10%) | 1 (13%) | 2 (11%) |
| **12 months post randomisation** | | | | |
| New diagnosis of epilepsy since discharge | n (%) | 0 (0%) | 1 (13%) | 1 (6%) |
| Antiepileptic treatment since discharge | n (%) | 0 (0%) | 0 (0%) | 0 (0%) |

ICU, intensive care unit.

recovery, defined as a LOS of >4. Ten participants (56%; IVIG n=5/10 (50%), placebo n=5/8 (63%)) had minor to severe sequelae. Table 4 displays the breakdown of these results.

Fourteen participants had a LOS assessment at 12 months after randomisation. Six participants (33%; IVIG n=4/10 (40%), placebo n=2/8 (25%)) had made full recovery at this timepoint, and eight participants (44%; IVIG n=4/10 (40%), placebo n=4/8 (50%)) had minor to severe sequelae.

### Paediatric Quality of Life Assessment
Seven participants (39%; IVIG n=5/10 (50%), placebo n=2/8 (25%)) had a PedsQL assessment at 4–8 weeks after discharge from acute care, and eight participants (44%; IVIG n=6/10 (60%), placebo n=2/8 (25%)) had a PedsQL assessment at 12 months after randomisation. At 4–8 weeks after discharge from acute care, the mean PedsQL score was 77.9 (SD 11.10) and 56.5 (SD 7.8) for the IVIG and placebo group, respectively. At 12 months, mean PedsQL scores were 79.9 (SD 21.6) and 63.7 (SD 30.1) for the IVIG and placebo groups, respectively. This data are displayed in table 4.

### Gross Motor and Function Classification System
Seven participants underwent a GMFCS assessment at 4–8 weeks after discharge from acute care, and eight participants underwent assessment at 12 months after randomisation. At 4–8 weeks after discharge, all seven participants assessed (39%; IVIG n=5/10 (50%); placebo n=2/8 (25%)) had mild impairment of gross motor functioning. At 12 months after randomisation, all eight participants (44%; IVIG n=6/10 (60%); placebo n=2/8 (25%)) experienced mild or severe impairment of gross motor function, as demonstrated by table 4.

### Strengths and difficulties assessment (SDQ)
SDQ results were available for seven participants (39%; IVIG n=5/10 (50%), placebo n=2/8 (25%)) at 4–8 weeks after discharge from acute care and eight participants (44%; IVIG n=6/10 (60%), placebo n=2/8 (25%)) at 12 months after randomisation.

At 4–8 week after discharge from acute care, five participants (28%; IVIG n=4 (40%); placebo n=1 (13%)) had a close to average SDQ score, one participant (6%; IVIG n=1/10 (10%)) had a slightly raised SDQ score and one participant (6%; placebo n=1/8 (13%)) had a very high SDQ score. At 12 months after randomisation, the same number of participants had a close to average score and slightly raised score, but two participants (11%; IVIG n=1/10 (10%), placebo n=1/8 (13%)) had a very high SDQ score.

### Adaptive Behaviour Assessment System—Second Edition (ABAS-II)
Eight participants had an ABAS-II assessment at 4–8 weeks after discharge from acute care, and seven participants had an ABAS-II assessment at 12 months after randomisation (see table 4). At 4–8 weeks after discharge, five participants (28%; IVIG n=4/10 (40%), placebo n=1/8 (13%)) had an ABAS-II score that was either similar or higher than the average score of the normative population, and three participants (17%; IVIG n=2/10 (20%), placebo n=1/8 (13%)) had a score that was lower than the average score. At 12 months after randomisation, the same number of participants had a score that was below the average at 12 months after randomisation, but four participants (22%; IVIG n=3/10 (30%), placebo n=1/8 (13%)) had a score that was either similar or higher than the average score at this timepoint.

### Neuropsychology outcomes
Thirteen participants (72%; IVIG n=8/10 (80%); placebo n=5/8 (63%)) had a neuropsychology assessment at 12 months after randomisation by a blided assessor. Four of these participants (30%; IVIG n=2/8 (25%), placebo

**Table 4** Secondary neurological and functional outcomes

| Outcome | 4–8 weeks post discharge | | 12 months post randomisation | |
| --- | --- | --- | --- | --- |
| | IVIG (n=10) | Placebo (n=8) | IVIG (n=10) | Placebo (n=8) |
| **LOS** | | | | |
| Severe sequelae | 2 (20%) | 2 (25%) | 2 (20%) | 2 (25%) |
| Moderate sequelae | 2 (20%) | 3 (38%) | 1 (10%) | 1 (13%) |
| Minor sequelae | 1 (10%) | 0 (0%) | 1 (10%) | 1 (3%) |
| Full recovery | 3 (30%) | 2 (25%) | 4 (40%) | 2 (25%) |
| Missing data due to withdrawal or loss to follow-up of participant | 1 (10%) | 1 (13%) | 2 (20%) | 2 (25%) |
| Missing data—assessment not performed | 1 (10%) | 0 (0%) | 0 (%) | 0 (%) |
| **PedsQL** | | | | |
| Mean (SD) | 77.9 (11.1) | 56.5 (7.8) | 79.9 (21.6) | 63.7 (30.1) |
| Missing data due to withdrawal or loss to follow-up of participant | 1 (10%) | 1 (13%) | 2 (20%) | 2 (25%) |
| Missing data—assessment not performed | 4 (40%) | 5 (63%) | 2 (20%) | 4 (50%) |
| **SDQ** | | | | |
| Close to average | 4 (40%) | 1 (13%) | 4 (40%) | 1 (13%) |
| Slightly raised | 1 (10%) | 0 (0%) | 1 (10%) | 0 (0%) |
| Very high | 0 (0%) | 1 (13%) | 1 (10%) | 1 (13%) |
| Missing data due to withdrawal or loss to follow-up of participant | 1 (10%) | 1 (13%) | 2 (20%) | 2 (25%) |
| Missing data—assessment not performed | 4 (40%) | 5 (63%) | 2 (20%) | 4 (50%) |
| **ABAS** | | | | |
| Very superior | 0 (0%) | 0 (0%) | 1 (10%) | 0 (0%) |
| Superior | 1 (10%) | 0 (0%) | 1 (10%) | 0 (0%) |
| Above average | 1 (10%) | 0 (0%) | 1 (10%) | 0 (0%) |
| Average | 2 (20%) | 1 (13%) | 0 (0%) | 1 (13%) |
| Below average | 0 (0%) | 1 (13%) | 1 (10%) | 0 (0%) |
| Borderline | 1 (10%) | 0 (0%) | 0 (0%) | 0 (0%) |
| Extremely low | 1 (10%) | 0 (0%) | 1 (10%) | 1 (13%) |
| Missing data due to withdrawal or loss to follow-up of participant | 1 (10%) | 1 (13%) | 2 (20%) | 2 (25%) |
| Missing data—assessment not performed | 3 (30%) | 5 (63%) | 3 (30%) | 4 (50%) |
| **GMFCS*** | | | | |
| Mild | 5 (50%) | 2 (25%) | 6 (60%) | 1 (13%) |
| Severe | | | 0 (0%) | 1 (13%) |
| Missing data due to withdrawal or loss to follow-up of participant | 1 (10%) | 1 (13%) | 2 (20%) | 2 (25%) |
| Missing data—assessment not performed | 4 (40%) | 5 (63%) | 2 (20%) | 4 (50%) |

| **GOSE-Peds at 6 months post randomisation** | | |
| --- | --- | --- |
| | IVIG (n=10) | Placebo (n=8) |
| Upper good recovery | 4 (40%) | 4 (50%) |
| Upper moderate disability | 1 (10%) | 1 (13%) |
| Upper severe disability | 0 (0%) | 1 (13%) |
| Lower severe disability | 3 (30%) | 1 (13%) |
| Missing data due to withdrawal or loss to follow-up of participant | 2 (20%) | 1 (13%) |
| Missing data—assessment not performed | 0 (0%) | 0 (0%) |

ABAS, Adaptive Behavior Assessment System; GMFCS, Gross Motor and Function Classification System; GOSE-Peds, Paediatric Glasgow Outcome Score Extended; LOS, Liverpool Outcome Score; SDQ, Strengths and Difficulty Questionnaire.

**Table 5** Neuropsychology outcomes at 12 months after randomisation

| Participant | Bayley cognitive score | FSIQ | VCI | VSI/PRI | WMI | PSI |
|---|---|---|---|---|---|---|
| Placebo arm | | | | | | |
| 1 | – | * | * | * | * | * |
| 2 | – | 79 | 95 | 79 | 75 | 71 |
| 3 | – | * | * | * | * | * |
| 4 | 110 | – | – | – | – | – |
| 5 | – | 89 | 99 | 88 | 83 | 94 |
| IVIG arm | | | | | | |
| 6 | – | * | * | * | * | * |
| 7 | – | 104 | 92 | 111 | 107 | 116 |
| 8 | – | 95 | 102 | 90 | 99 | 91 |
| 9 | – | 88 | 93 | 96 | 91 | 83 |
| 10 | 55 | – | – | – | – | – |
| 11 | – | 65 | 60 | 75 | 72 | – |
| 12 | – | * | * | * | * | * |
| 13 | – | 119 | 108 | 110 | 110 | 131 |

Key: green = normal neurodevelopmental score, yellow = mild impairment, red = severe impairment.
*Young person unable to complete full battery due to attention or behavioural needs
FSIQ, full-scale IQ; PRI, perceptual reasoning index; VCI, verbal comprehension index; VSI, visual spatial index; WMI, working memory index.

n=2/5 (40%)) were unable to complete the full battery of assessments due to attentional or behavioural needs.

Five participants (28%; IVIG n=4/10 (40%), placebo n=1/8 (13%)) had a score of ≥85 (indicating normal development) for full-scale IQ (FSIQ), six (33%; IVIG n=4/10 (40%); placebo n=2/8 (25%)) for verbal comprehension, five (28%; IVIG n=4/10 (40%), placebo n=1/8 (13%)) for visual spatial; four (22%; IVIG n=4/10 (40%)) for working memory; and four (22%; IVIG n=3/10 (30%); placebo n=1/8 (13%)) for perceptual reasoning (PRI). Two participants (IVIG n=1, placebo n=1) were assessed using the Bayley scale of infant development, one participant (IVIG n=1) had severe neurodevelopmental impairment while the other (placebo n=1) had a normal neurodevelopmental outcome. These results are displayed in table 5.

### Neuroimaging outcomes
Nineteen acute neuroimaging scans were available for 13 participants (72%; IVIG n=8/10 (80%), placebo n=5/8 (63%)). Five of these scans (for five unique participants; IVIG n=2/8 (25%), placebo n=3/8 (38%)) had abnormal findings; all of these were MRI scans (see online supplemental table 1). Four of the abnormal scans showed bilateral lesions.

There were nine follow-up scans for eight unique participants (IVIG n=5/10 (50%), placebo n=3/8 (50%)); six of these scans (for five unique participants; IVIG n=3/5 (60%), placebo n=1/4 (25%)) were normal and unchanged from the acute scan. Three follow-up scans (for three unique participants; IVIG n=2/5 (40%), placebo n=1/3 (33%)) had abnormal findings; two of these were unchanged from the acute scans and an acute scan was not available for comparison one participant.

### Autoantibody testing
Twelve participants (67%; IVIG n=7/10 (70%), placebo n=5/8 (63%)) had autoantibody testing. One participant (placebo n=1) was positive for LGI1 antibodies, and one participant (placebo n=1) was positive for MOG antibodies. Two additional participants (IVIG n=2) were positive for IgG binding to the surface of live neurons, indicating the presence of IgG antibodies binding to neurons, but negative for antibodies to the specific antigens tested, indicating the presence of undefined IgG antibodies that could be pathogenic.

### Safety data
Ten serious AEs occurred in three participants in the placebo group and none in the IVIG group. None of the SAEs were judged to be related to the study treatment. One participant in the IVIG group reported an AE of special interest; the participant developed a fever during the IVIG infusion; however, this was judged to be unrelated to the study treatment. None of the participants experienced haemolysis following receipt of two doses of study treatment. No deaths occurred during the study period.

### DISCUSSION
The IgNiTE study was terminated early due to slower than expected recruitment and was therefore unable to provide conclusive evidence regarding the efficacy of

IVIG in the treatment of childhood encephalitis. Thus, it remains unknown whether early administration of IVIG in children with all-cause encephalitis offers clinical benefit.

While the IgNITE study was unable to address the primary study objective, the results do provide evidence of the poor outcomes experiences by many children with encephalitis. Almost a third of participants made a poor recovery based on GOS-E Peds assessment at 12 months after randomisation. Other measures of neurological outcomes consistently demonstrated a heavy burden of disability; 44% of patients had minor to severe sequalae at 12 months according to the LOS assessment, and the same proportion of patients experienced mild or severe impairment of gross motor function at the same time-point. The proportion of children with functional impairments on the SDQ and ABAS-II assessments at 12 months after randomisation was lower, but this was likely due to fewer participants completing these assessments.

The results also demonstrate the impact of childhood encephalitis on healthcare systems. Over half of participants required admission to intensive care during the acute illness, and 90% of these children were intubated. The overall median length of acute hospital care for participants was 11 days, compared with a mean length of hospital stay of 1.64 days for children and young people following an emergency admission in the UK.[35] Furthermore, given the high proportion of participants with lasting disability, many children with encephalitis are likely require ongoing non-acute hospital care for neurorehabilitation.

These data are consistent with previous studies of childhood encephalitis in high-income settings. In a prospective Australian study involving 287 children with encephalitis, 49% of children required admission to intensive care, median length of hospitalisation was 11 days and 27% of children had moderate to severe neurodisability at hospital discharge.[4] Of note, they used the adult Glasgow Outcome Score tool for assessment of outcomes and did not capture children with mild-to-moderate neurodisability, which may explain the lower proportion of children with reported neurodisability compared with the IgNiTE study. A meta-analysis evaluating long-term outcomes of childhood encephalitis reported 47% of children to have long-term sequalae in studies in high-income countries, although there was no standardised definition of sequalae used across these studies.[17]

### Limitations of the study

The main limitation of the IgNiTE study is that the predefined sample size was not met, and the primary study objective was therefore not achieved. The study initially planned to recruit 308 participants over a 24-month period. The sample size calculation was based on the anticipated number of annual encephalitis hospital admissions in the UK and the anticipated treatment effect of IVIG, based on a large observational study on autoimmune encephalitis.[26 36] However, recruitment to the study was slower than expected. Of the 884 children assessed for eligibility, 63% (561) were excluded because they did not meet the case definition for encephalitis, suggesting that the use of strict diagnostic criteria may have precluded the inclusion of some children with clinically suspected encephalitis. A further 12.5% were excluded due to insufficient clinical results being available to satisfy the eligibility criteria within the time frame for participant enrolment. The initial screening form used did not capture the reason for exclusion; hence, this was not recorded for the first 10% of children assessed for eligibility.

Overall, 13% (115) of children were assessed to meet to inclusion criteria, but 55% (63) of these children fulfilled exclusion criteria and were thus ineligible. The main reasons for exclusion were prior or planned IVIG treatment as part of routine care (32%), and study timeline restrictions (24%). The use of IVIG as part of routine care demonstrates that some clinicians were already convinced of the benefit of IVIG in childhood encephalitis despite the lack of high-quality evidence and the fact that at the time the trial was undertaken, IVIG was not commissioned for routine use in acute childhood encephalitis. This highlights the importance of ensuring that there is equipoise among treating clinicians when conducting randomised controlled trials.

Recruitment to the trial was also impacted by a lower than anticipated consent rate. Of the 52 children who were eligible for enrolment, participation was declined in 65% of cases. This is not unexpected given the requirement for parents or guardians to provide informed consent at an exquisitely sensitive time for the family. Other factors which may have contributed to the low consent rate include the limited time frame for enrolment and the trial duration.[37]

Finally, recruitment was impacted by delays in the participating NHS hospitals opening as recruitment sites, due primarily to shortages of research personnel and delays in local approval processes. Nine of the 21 participating hospitals did not recruit any particitpants during the study; 5 of these hospitals were open to recruitment for 6 months or less.

### Lessons learned and future research

Further research is required to establish whether early IVIG is of therapeutic benefit in the treatment of childhood encephalitis, irrespective of the underlying aetiology. The IgNiTE study demonstrated the feasibility of conducting a randomised controlled trial to investigate this important question. Future studies should anticipate the recruitment challenges discussed above and consider strategies such as incorporating a pilot phase, using less strict entry criteria, allowing a wider time frame in which participants can be enrolled, and adopting approaches to optimise consent rates in eligible patients.

### Conclusion

The IgNiTE study was terminated prematurely due to slow recruitment and therefore did not reach the

predetermined sample size required to evaluate the effect of IVIG compared with placebo in childhood encephalitis. However, the study results support existing evidence of poor neurological outcomes in many children with encephalitis. This provides further compelling evidence of the need for better treatments in childhood encephalitis. Future studies are required to establish if treatment with IVIG is of benefit in children with encephalitis of all causes. Such studies should take into account the challenges encountered and lessons learnt from the IgNiTE study.

**Author affiliations**
[1]Oxford Vaccine Group, Department of Paediatrics, University of Oxford, Oxford, UK
[2]Department of Paediatrics, University of Oxford, Oxford, UK
[3]Vaccine Evaluation Center, BC Children's Hospital Research Institute, University of British Columbia, Vancouver, British Columbia, Canada
[4]Department of Pediatrics, University of British Columbia, Vancouver, British Columbia, Canada
[5]Children's Neurosciences, Evelina London Children's Hospital Neurosciences Department, London, UK
[6]Department of Womens and Childrens Health, Faculty of Life Sciences and Medicine, King's College London, London, UK
[7]UCL Great Ormond Street Institute of Child Health, London, UK
[8]The Encephalitis Society, Malton, UK
[9]Department of Clinical Infection, Microbiology and Immunology, University of Liverpool, Liverpool, UK
[10]Clinical Health Psychology, Alder Hey Children's NHS Foundation Trust, Liverpool, UK
[11]Department of Neurology, Alder Hey Children's NHS Foundation Trust, Liverpool, UK
[12]Institute of Infection, Veterinary and Ecological Sciences, University of Liverpool, Liverpool, UK
[13]Department of Paediatric Neurology, Oxford University Hospitals NHS Trust, Oxford, UK
[14]National Institute for Health Research Health Protection Research Unit in Emerging Zoonotic Infections, University of Liverpool, Liverpool, UK
[15]Walton Centre NHS Foundation Trust, Liverpool, UK
[16]The Pandemic Institute, Liverpool, UK
[17]Nuffield Department of Clinical Neurosciences, University of Oxford, Oxford, UK
[18]Weatherall Institute of Molecular Medicine, University of Oxford, Oxford, UK
[19]Nuffield Department of Primary Care Health Sciences, University of Oxford, Oxford, UK

**Acknowledgements** The authors would like to thank all the participants, families, and health-care professionals for their involvement in the study.

**Collaborators** The IgNiTE Study Team: Trial Management Group: Prof Sir Andrew J Pollard (Chief Investigator), Dr Ming Lim, Prof Tom Solomon, Dr Rachel Kneen, Dr Michael Absoud, Dr Mike Pike, Dr Manish Sadarangani, Dr Kling Chong, Dr Chris A. Clark, Dr Victoria Gray. Trial coordinating team (Oxford): Dr Mildred A Iro (Clinical ResearchFfellow), Louise Willis (Research Nurse/Project Manager), David Kerr, Sophie Bradshaw, Svetlana Milca, Simon Kerridge, Emma Plested, Yama Mujadidi. IMP management: Shakeel Herwitker, Mandy Wan. Trial Steering Committee: Dr Claire Cameron, Dr Ming Lim, Dr Adilia Warris, Dr Federico Martinon-Torres, Mike Bale, Sonia Bale, Alan Percival, and Dr Ava Easton. Data Monitoring and Ethics Committee: Dr Charles Warlow (Chairperson), Dr Jo Haviland (statistician), Dr David Pace and Dr Simon Nadel. Statistics team: Dr Ly-Mee Yu, Dr Meryn Voysey, Dr Xinxue Liu, Liberty Cantrell. Laboratory team (Oxford): Sagida Bibi, Amy Beveridge, Amber Thompson, Daniel O'Connor, Prof Sarosh Irani, Dr Patrick Waters. Neuropsychological assessment support: Lauren Burke, Victoria Gray. Oxford Vaccine Group: Emma Plested, Parvinder Alley. Recruitment sites: Principal Investigators: Andrew Pollard, Oxford University Hospitals, Sanjay Bhate, Great Ormond Street Hospital, Rachel Kneen, Alder Hey Children's Hospital, Paul Heath, St George's Hospital, Anna Riddell, Barts and the London (Royal London), Ming Lim, Guy's and St Thomas's - Evelina, Archana Desurkar, Sheffield Children's Hospital, Elma Stephen, NHS Grampian, Steve Welch, Heart of England, Paddy McMaster, Pennine - North Manchester Children's Hospital, Alice Jollands, Ninewells (Tayside Health Board), Jay Shetty, Royal Hospital for Sick Children, Edinburgh, Kayal Vijayakumar, University Hospitals Bristol, Leena Mewasingh, Imperial – St Mary's Hospital, William Whitehouse, Nottingham University Hospitals, Vishal Mehta, Hull Royal Infirmary, John Alexander, University Hospitals of North Midland, John Livingston, Leeds Teaching Hospital, Christian de Goede, Royal Preston Hospital, Dominic Smith, York Teaching Hospital, Andrew Collinson, Royal Cornwall Hospitals.

**Contributors** The trial was conceptualised by MP and AJP, and designed with input from ML, MA, MS, MI, L-MY, RK and TS. MP, MS, RK, TS, WKC, CC, ML, MA, AV, VG, AE, L-MY and AJP are named investigators on the IgNiTE trial. Each coauthor provided specific additional contributions within their area of expertise; paediatric neurology (MP, RK, ML, MA), paediatric infectious diseases (MS, AJP), paediatric neuropsychology (VG), neuroimaging (CAC, WKC), neuroimmunology (AV), statistics (L-MY, LC, XL) and patient group (AE). MI is the lead doctor for the trial and LW is the lead nurse for the trial. MH prepared the first version of the manuscript. All authors contributed to the manuscript and have approved the final manuscript for publication. AJP is the overall guarantor.

**Funding** This project was funded by the National Institute for Health Research (NIHR) Efficacy and Mechanism Evaluation programme (grant number 12/212/15). The investigational medicinal product (IMP), intravenous immunoglobulin (Privigen) was provided by CSL Behring.

**Competing interests** MI was a trainee member on the NIHR Efficacy and Mechanism Evaluation Programme Funding Committee from October 2020 to October 2021. MA has received a grant from the NIHR in the last 36 months, for research unrelated to the submitted work. MS has been an investigator on projects funded by GlaxoSmithKline, Merck, Moderna, Pfizer, Sanofi-Pasteur, Seqirus, Symvivo and VBI Vaccines; all funds have been paid to his institute. Ava Easton is Chief Executive of the Encephalitis Society, which has previously received grants from CSL Behring (UK). Ming Lim has received grants from the GOSH charity, Boston Children's Hospital Research Fund and Action Medical Research in the last 36 months, all for research unrelated to the submitted work. Ming Lim is co-chair of the European Paediatric Neurology Education and Training Board and works for an institution which holds research accounts with Roche (Switzerland), Octapharma (Switzerland) and Novartis (Switzerland). Tom Solomon is supported by the NIHR Health Protection Research Unit in Emerging and Zoonotic Infections, NIHR Programme Grant for Applied Research, NIHR Global Health Research on Brain Infections and the European Union's Horizon 2020 research and innovation program ZikaPLAN. Tom Solomon is a consultant for the MHRA Vaccine Benefit Risk Expert Working Group. Angela Vincent is a consultant for Aspen NewCo Inc and has received honoraria from UCB and Alexion. L-MY was a member of the NIHR Health Technology Assessment Efficient Study Designs from November 2015 to July 2016. Andrew J Pollard is chair of the Department of Health and Social Care's Joint committee on Vaccines and Immunisation (JCVI) and was a member of WHO's SAGE until 1 January 2022. Oxford University has entered a partnership with AstraZenenca on COVID-19 vaccines, but Andrew Pollard does not participate in the JCVI COVID-19 committee.

**Patient and public involvement** Patients and/or the public were involved in the design, or conduct, or reporting, or dissemination plans of this research. Refer to the Methods section for further details.

**Patient consent for publication** Not applicable.

**Ethics approval** The study was approved by the UK National Research Ethics Service (NRES) committee (South Central—Oxford A; REC 14/SC/1416). Written approval from local Research and Development (R&D) departments at each participating site was obtained before recruitment was commenced at each site. This study was conducted in accordance with the Declaration of Helsinki (1996), in full conformity with the International Conference on Harmonisation of Technical Requirements for Registration of Pharmaceuticals for Human Use (ICH) Guidelines for Good Clinical Practice (CPMP/ICH/135/95 July 1996), the Research Governance Framework, and the Medicines for Human Use (Clinical Trial) Regulations 2004.

**Provenance and peer review** Not commissioned; externally peer reviewed.

**Data availability statement** All data requests should be submitted to the corresponding author for consideration. Access to an anonymised dataset may be granted following review.

of the translations (including but not limited to local regulations, clinical guidelines, terminology, drug names and drug dosages), and is not responsible for any error and/or omissions arising from translation and adaptation or otherwise.

**ORCID iDs**
Matilda Hill http://orcid.org/0000-0002-1566-2512
Xinxue Liu http://orcid.org/0000-0003-1107-0365
Andrew J Pollard http://orcid.org/0000-0001-7361-719X

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
