## [Reviewer comments · BMJ Open]

ARTICLE DETAILS

TITLE (PROVISIONAL)	Intravenous immunoglobulin treatment in childhood encephalitis (IgNiTE): A randomised controlled trial
AUTHORS	Hill, Matilda; Iro, Mildred; Sadarangani, Manish; Absoud, Michael; Cantrell, Liberty; Chong, Kling; Clark, Christopher; Easton, Ava; Gray, Victoria; Kneen, Rachel; Lim, Ming; Liu, Xinxue; Pike, Michael; Solomon, Tom; Vincent, Angela; Willis, Louise; Yu, Ly-Mee; Pollard, Andrew; The, IgNiTE study team

VERSION 1 – REVIEW

REVIEWER	Perna, Annalisa Mario Negri Institute, Renal Medicine
REVIEW RETURNED	18-Apr-2023

GENERAL COMMENTS	Summary The authors randomized 18 children aged 6 months to 16 years with diagnosis of acute or sub-acute encephalitis to intravenous immunoglobulin (IVIG) treatment or matching placebo (IgNiTE trial). They studied the 'good recovery' at one year - defined as score <2 on the Paediatric Glasgow Outcome Score Extended (GOS-E peds)- as primary outcome. Secondary outcomes were clinical, neurological, neuroimaging and neuropsychological results, proportion of children with immune-mediated encephalitis and IVIG safety data. This randomised, double-blind, placebo-controlled trial did not reach the planned sample size of 308 participants due to slower than expected recruitment, resulting in early trial termination. Then the Authors decided to provide descriptive results only. This is an important area of research in view of the public health importance of the topic. Without any doubt there are several interesting findings in the manuscript, including to what extent it is difficult to perform a trial in this area of research. Some comments listed below are intended to be constructive and helpful to design future studies similar to IgNiTE trial. Comments According to Study Protocol interim reports periodically reviewed by a Data and Safety Monitoring Committee were planned (page 50). It is however unclear whether formal statistical stopping rules were applied. It would be useful to know which rules for stopping the trial early for efficacy/harm/futility were foreseen, including number and timing for the planned interim analyses.
---

	Among the reasons for exclusion after eligibility assessment the Authors listed 'insufficient information to assess eligibility (n=111, 13%) and 'reason for exclusion unknown' (n=95, 11%) (Figure 1). Because the above two reasons account for a non-negligible proportion of excluded participants the Authors should further comment on this point. There were 21 participating centres randomising a total of 18 children (i.e. less than one child per centre). The proportion of excluded/withheld consent was very high (excluded 832, 94%, withheld after eligibility 34, 65%). Were the above proportions similar across all participating centres? Did some centres reach the target of children to be randomised? There is a discrepancy between the sample size mentioned in the manuscript (Statistical Analysis, page 6: total n=308, 20% difference, 90% power, 10% dropout) and the number of participants specified in the protocol (par 11.2, page 50): total n=154, 25% difference, 80% power, 10% dropout. Please clarify. Please, add the planned sample size in the Abstract. Please, add median [range] follow up in the Results section.
--	---

REVIEWER	Mizuguchi, Masashi National Rehabilitation Center for Children with Disabilities, Pediatrics
REVIEW RETURNED	21-Apr-2023

GENERAL COMMENTS	This article by Hill et al gave a detailed account of the failure of a study to evaluate the efficacy of intravenous immunoglobulin (IVIG) treatment in childhood encephalitis. The study was originally designed as a multi-center double-blind, randomized placebo-controlled trial, but was terminated early due to slow recruitment. Of a total of 884 patients assessed for eligibility, only 18 participated and were given either IVIG (n=10) or placebo (n=80). No statistical analysis was performed. There were no data suggesting the efficacy of IVIG. When published, the readers of this article would be most interested in the reasons for the failure, since they may provide lessons for future studies. In this regard, the most important results are Figure 1, rather than Tables 1-5.  1. Which inclusion criteria were not met by many patients? What was the insufficient information to assess eligibility? What are the suspected reasons for patients with "reason for exclusion unknown"? 2. "Limitations of the study" should be discussed in much more detail, and in a more quantitative fashion. 3. If possible, the final diagnosis of encephalitis (e.g. viral, autoimmune and unclassified) in the 18 patients should be shown.
---

VERSION 1 – AUTHOR RESPONSE

Reviewer 1 comments	Authors reply
According to Study Protocol interim reports periodically reviewed by a Data and Safety Monitoring Committee were planned (page 50). It is however unclear whether formal statistical stopping rules were applied. It would be useful to know which rules for stopping the trial early for efficacy/harm/futility were foreseen, including number and timing for the planned interim analyses.	The trial was stopped early due to withdrawal of funding due to slower than anticipated recruitment (explained page 5 of main text) and not because the stopping guidance criteria were met. The stopping guidelines are available in the published protocol paper (as described page 4 of main text and referenced – see bottom of this section). The planned interim analyses are specified in the DSMC charter and statistical analysis plan (both of which we would be happy to share). The DSMC members had access to safety data on demand, and the original plan was for a formal safety report to be analysed after the first 50 patients had been randomized; the frequency of further analyses would be determined at this point. In the event, only 18 participants were randomised before the trial was halted. Results from the interim analyses of the accrued data were assessed prior to the trial being halted. This information is not provided in full in the submitted paper as it is not deemed relevant to the interpretation of results. “This trial may be suspended or prematurely terminated by the sponsor, CI, regulatory authority or funder if there is sufficient reason to think that the safety of participants is affected by the trial procedures. Written notification, documenting the reason for trial suspension or termination, will be provided by the suspending or terminating party to the investigator, funders and regulatory authorities. If the trial is prematurely terminated or suspended, the CI will promptly inform the REC, MHRA and CSL Behring and will provide the reason(s) for the termination or suspension.” Page 11 of Iro MA et al. Immunoglobulin in the Treatment of Encephalitis (IgNiTE): protocol for a multicentre randomised controlled trial. BMJ Open. 2016 Nov 3;6(11):e012356. doi: 10.1136/bmjopen-2016-012356. PMID: 27810972; PMCID: PMC5129051.

Among the reasons for exclusion after eligibility assessment the Authors listed 'insufficient information to assess eligibility (n=111, 13%) and 'reason for exclusion unknown' (n=95, 11%) (Figure 1). Because the above two reasons account for a non-negligible proportion of excluded participants the Authors should further comment on this point.	The initial screening form used to assess eligibility did not capture why children were excluded, hence this information was not recorded for the first 95 children excluded. The form was subsequently modified to enable this information to be captured. 111 children were excluded due to insufficient clinical results being available to satisfy the eligibility criteria within the time frame for participant enrolment. We have added a discussion of this to the 'Limitations of the study' section on page 13 of the main text.
There were 21 participating centres randomising a total of 18 children (i.e. less than one child per centre). The proportion of excluded/withheld consent was very high (excluded 832, 94%, withheld after eligibility 34, 65%). Were the above proportions similar across all participating centres? Did some centres reach the target of children to be randomised?	None of the centres reached their recruitment target. The recruitment target for each site was based on the size of the hospital (1-2 cases expected per year for DGH & 8-10 cases per year in large teaching hospitals, based on the paper at the bottom of this section – cited as ref.37 in main text), assuming a consent rate of 30% and extrapolated based on the duration it had been open as a recruitment site (not all centres opened to recruitment simultaneously). At the time of the study closing, 9 centres had not recruited any participants, and 12 centres had recruited 1-3 participants (25-40% of their recruitment targets based on the time they had been open for). Most of hospitals which did not recruit any patients were open for 6 months or less. We have added some of this detail and analysis to the 'Participants' section of the Results and as a final paragraph in the 'Limitations of the study' (page 13) section of the main paper. We have also added further information regarding the exclusion of participants to the 'Limitations of the study' (page 13) section of the main paper. Kneen R, Michael BD, Menson E, Mehta B, Easton A, Hemingway C, et al. Management of suspected viral encephalitis in children - Association of British Neurologists and British Paediatric Allergy, Immunology and Infection Group national guidelines. J Infect. 2012;64(5):449-77.
There is a discrepancy between the sample size mentioned in the manuscript (Statistical Analysis, page 6: total n=308, 20% difference,	Many thanks for pointing out this discrepancy, which is caused by an error in the most recent

90% power, 10% dropout) and the number of participants specified in the protocol (par 11.2, page 50): total n=154, 25% difference, 80% power, 10% dropout. Please clarify.	version of the protocol (Protocol 7.0, dated 22 March 2022). The original protocol (published 03 November 2016, as stated page 4 of main text) calculated a sample size of 308 based on the calculations described. This is the target sample size that was used throughout the trial whilst running. After the trial was halted (due to withdrawal of funding due to slower than anticipated recruitment) a proposal was put forward to reduce the sample size to 154. Despite this suggestion, funding was not reinstated, and the original sample size was therefore not altered. However, a revised sample size of 154 was erroneously included in section 11.2 'The Number of Participants' in the most recent protocol amendment, which was completed after the study had closed (Protocol 7.0, dated 22 March 2022). Of note, a sample size of 308 is correctly cited throughout the rest of this protocol. A file note has been made to explain this error, and we are happy to share this with the reviewers. We have therefore referred to the correct planned sample size of 308 throughout the text of the manuscript.
Please, add the planned sample size in the Abstract	We have added this to the abstract
Please, add median [range] follow up in the Results section	Follow up was performed at the intervals specified. Please clarify if additional information required.

Reviewer 2 comments	Authors reply
Which inclusion criteria were not met by many patients? What was the insufficient information to assess eligibility? What are the suspected reasons for patients with "reason for exclusion unknown"?	We have revised the 'Limitations of the study' (page 13) to discuss the reasons patients were excluded in more detail.
"Limitations of the study" should be discussed in much more detail, and in a more quantitative fashion.	As above, we have revised the 'Limitations of the study' (page 13) to provide further information, including quantitative detail.
If possible, the final diagnosis of encephalitis (e.g. viral, autoimmune and unclassified) in the 18 patients should be shown.	We have described the results of the autoantibody testing (page 12 of main text) – 2 participants had specific antibodies identified which were diagnostic of autoimmune encephalitis, and a further 2 participants had antibody binding to live neurons detected,

	indicative of possible pathogenic, undefined antibodies. We do not have the final diagnosis for all other participants. A secondary objective of the trial was to ascertain the proportion of participants with immune-mediated encephalitis; however, it did not set out to describe the cause for the encephalitis in all patients, and thus the data are unfortunately not available.
--	--

VERSION 2 – REVIEW

REVIEWER	Perna, Annalisa Mario Negri Institute, Renal Medicine
REVIEW RETURNED	18-Aug-2023

GENERAL COMMENTS	The Authors properly addressed previously arisen issues.
--

REVIEWER	Mizuguchi, Masashi National Rehabilitation Center for Children with Disabilities, Pediatrics
REVIEW RETURNED	26-Jul-2023

GENERAL COMMENTS	In this revised version, authors have well addressed most of the issues in the original version.
--